# Physiological Parameters to Identify Suitable Blood Donor Cows for Preparation of Platelet Rich Plasma

**DOI:** 10.3390/ani11082296

**Published:** 2021-08-04

**Authors:** Anna Lange-Consiglio, Rosangela Garlappi, Chiara Spelta, Antonella Idda, Stefano Comazzi, Rita Rizzi, Fausto Cremonesi

**Affiliations:** 1Department of Veterinary Medicine, Università degli Studi di Milano, 26900 Lodi, Italy; antonella.ida@yahoo.it (A.I.); stefano.comazzi@unimi.it (S.C.); rita.rizzi@unimi.it (R.R.); fausto.cremonesi@unimi.it (F.C.); 2Independent Researcher, 20133 Milan, Italy; rosgarlappi@alice.it; 3Independent Researcher, 27100 Pavia, Italy; chspelta@icloud.com

**Keywords:** cow, blood donor, mastitis, PRP, yield

## Abstract

**Simple Summary:**

Platelet rich plasma is a biological product obtained from blood and used for regenerative treatments of different pathologies. It is characterized by a high concentration of platelets (at least 3 times the physiological level) containing many growth factors with anti-inflammatory, bactericidal and regenerative properties. In human medicine, PRP is used in an autologous way, it means that the blood donor is also the recipient. In veterinary medicine, PRP is used to treat different diseases or lesions and in bovine species to treat mastitis. In this context, the opportunity to have PRP ready to use, stored from donor cows of the same farm where it will be used, would be very useful in treating this pathology immediately when it occurs. For this purpose, the present research aimed to detect parameters useful to identify the most suitable cows to be used as blood donors to obtain the highest yield of PRP (milliliters of PRP obtained with respect to milliliters of initial blood). Our results showed that blood collection from the mammary vein within three months of parturition, from nonpregnant cows at 5 years of age, but not the blood collection season, were associated with a high yield of PRP.

**Abstract:**

Platelet rich plasma (PRP) has been shown to be beneficial in the treatment of bovine mastitis, with an action comparable to that of antibiotics. Autologous treatment is feasible in experimental conditions but is difficult to apply in field conditions, particularly in acute mastitis. The ideal scenario would be to have heterologous PRP stored on every farm so that it is readily available when needed. In this paper, we analysed data collected during bovine mastitis treatment with heterologous PRP produced by casual donor cows on several farms. We tried to identify parameters which might be useful to identify the most suitable cows to be used as blood donors, to obtain the highest yield of PRP. Variables considered for each animal were the age, the parity, the date of the last parturition, the season of blood collection, the site of blood collection (jugular or mammary vein) and the reproductive status e.g., pregnant or not pregnant. There were statistically significant differences for all the variables considered from the 135 blood cows, except for the blood collection season. The highest yield of PRP was associated with nonpregnancy blood collection within three months of parturition, parity 3 or 4, and blood collection from the mammary vein.

## 1. Introduction

Platelet rich plasma has been defined as plasma that contains a platelet concentration above the ‘‘normal’’ physiologic level found in whole blood. Platelets hold about 50–80 *α*-granules that contain hundreds of bioactive proteins, including a wide range of growth factors [1]. The most important growth factors in this context are platelet-derived growth factor (PDGF), vascular endothelial growth factor (VEGF), fibroblast growth factor (FGF), transforming growth factor-beta 1 (TGF-β1), epidermal growth factor (EGF), insulin-like growth factor (IGF), connective tissue growth factor (CTGF), and hepatocyte growth factor (HGF) [2].

An increased concentration of platelets yields an increase in the concentration of growth factors that are stored in the α-granules of platelets. Numerous in vitro studies have shown a direct dose–response influence of many growth factors on cell migration, cell proliferation and matrix synthesis [3,4,5,6]. In this context, it has been proposed that the local administration of increased concentrations of these growth factors using PRP could optimize the local healing environment and thus enhance the ability of pathologically compromised tissues to generate a repair response [7].

In human medicine, PRP is used in vivo for the treatment of degenerative pathologies of the knee, hip, ankle and hyaline cartilage due to degenerative or traumatic injuries [8,9]. Platelet rich plasma has been shown to have a longer lasting effect than infiltration with hyaluronic acid or corticosteroids. In human orthopedics, promising results have also been found in young sportsmen with acute and chronic tendinopathies and muscle injuries. In human dermatology, PRP is mainly used to counteract skin aging on the face, since PRP stimulates the production of collagen and hyaluronic acid and, recently, it has also been used in diabetic patients for the treatment of skin ulcers. Platelet rich plasma is also used in maxillofacial surgery, dentistry, dermatology, cosmetic surgery, orthopaedics, and sports medicine [10,11,12,13,14]. Platelet rich plasma has anti-inflammatory [15,16,17], bactericidal and regenerative properties [8] and wound healing is promoted by the natural growth factors and cytokines contained in the platelets [9].

In the veterinary field, enriched plasma has been mainly used to treat deep wounds, chronic wounds, skin ulcers and fistulas in the dog [18] and to aid healing of intestinal disease in pigs [11]. In sport horses, PRP is used to treat tendinopathies [19] and in bovines to treat mastitis [20]. Usually, PRP is used in an autologous way, but in the treatment of mastitis, due to the large number of animals under treatment, a large amount of pre-prepared PRP is needed to treat acute mastitis. If PRP is stored from donor cows on the farm where it is to be used, it will be readily available when needed.

During the preparation of heterologous PRP for mastitis treatments, it is understood that the yield of PRP (milliliters of PRP obtained with respect to milliliters of initial blood) is very different between individual animals. Therefore, it would be helpful to identify parameters that might aid to identify an ideal blood donor, with a good yield of PRP.

To date, in veterinary medicine, the parameters for identifying the most suitable donors have yet to be defined. In human medicine, by Italian law, the requirements for the blood donor are clear and regulated by the Ministerial Decree of 2 November 2015 [21] “Provisions relating to the quality and safety requirements of blood and blood components’’. This Decree establishes the physical requirements of a human blood donor. In addition, it establishes that the number of donations per year must not exceed four in men and two in women. Moreover, a human blood donor must also undergo periodic checks and mandatory tests, such as the complete blood count, tests for the biological qualification of the blood and blood components, in particular a serological test for anti-HIV 1–2 antibodies and HIV 1–2 antigens. People affected by important pathologies such as autoimmune diseases, infectious diseases, celiac disease, cardiovascular diseases, hypertension, epilepsy, neoplasms and diabetes, cannot be donors.

In the present study, data collected in the treatment of mastitis with PRP [20] and data collected within an experimental setting funded by the European Union project (PSR 2014–2020 Decree No. 9243 of 27 July 2017 compart 1.2.01 “Demonstration projects and information actions” of the 2014/2020 Rural Development Program of Lombardy called MASTOP, ID 201600553231), were retrospectively analyzed. These analyses were used to establish whether a relationship between the yield of PRP with respect to the initial blood volume, and the health status as well as the housing conditions of the donor cows, was apparent.

Consequently, the aim of this study was to identify, through the evaluation of farm parameters and physiological data related to individual animals, which factors impact PRP yield, and which features indicate that a cow will be a good PRP donor.

## 2. Materials and Methods

Written informed consent from the owners was obtained to allow blood collection in cows.

### 2.1. Breeding Identification

Forty farms located around Lodi (northern Italy) were selected. These had some characteristics in common such as the breed of cows, type and characteristics of housing, feeding system, degree of hygiene of the animals, their body condition score (BCS), and nutritional status. All cows were the Friesian breed, housed in stalls with no overcrowding (each animal had a cubicle yard) and with a bedding consisting mainly of straw and manure. All animals were fed with a total mixed ration, hygiene related to the milking method was good/excellent on all farms and the body condition score (BCS) of each animal was found to be good (2.75–3.50 depending on the physiological stage) [22]. Only farms equipped with cooling systems consisting of fans and ceiling blades, and with hand showers and nebulizers were selected, since blood samples were collected mainly during the summer season during the months of July, August, and September with temperatures above 30 °C.

### 2.2. Bovine Identification

All farms were free from transmissible infectious diseases such as bovine rhinotracheitis (IBR), bovine viral diarrhea (BVD) and paratuberculosis (by serological tests) and were checked for their vaccination plans.

Each farm made available from 2 to 4 cows, for a total of 135 cows.

The 135 cows enrolled in this study were selected by the breeder and the veterinary surgeon based on their nutritional status, the absence of pathologies and good general animal welfare. The other parameters recorded for each animal were the physiological status (pregnant or not pregnant), the season of blood collection (autumn or summer), the date of the last parturition (interval from last delivery to time of blood collection), the site of blood collection (jugular or mammary vein), the age and number of births.

Regarding the presence or absence of pregnancy, all the cows were lactating and 75.6% (102 cows) were not pregnant.

In 62.2% (84 cows) of cows, blood was collected in the summer (21 June–21 September), while in the remaining 37.8% (51 cows) blood was collected in the autumn (22 September–21 December).

The cows were distributed with 55.55% (75 cows) in the interval <3 months from parturition, 16.30% (22 cows) in the interval between 4 and 5 months and 28.15% (38 cows) in the interval >6 months.

Blood samples were collected from two different sites. Only 11% (15 cows) of the samples were taken from the jugular vein and the remaining 89% (120 cows) from the mammary vein.

In regard to the age, one-third (33.3%; 45 cows) of the cows were between 24 and 48 months of age, 48.9% (66 cows) were between 49 and 72 months of age, 13.3% (18 cows) were 73–89 months of age and 4.4% (6 cows) were 97–120 months of age.

Of the cows in the study, 12 were primiparous, while the other 123 were pluriparous.

The platelet count (PLT) in peripheral blood was also recorded. The randomization in the selection of the donor cows meant a lack of standardization for the parameters analyzed, such that there was not an equal number of subjects in each subcategory. This generated a disproportion in the statistical analysis.

#### 2.2.1. PRP Preparation

##### Collection of Blood

After sterile preparation of a few centimetres of skin, a 16-gauge butterfly catheter (Terumo Europe NV, Leuven, Belgium) was used to take 450 mL of blood from the subcutaneous mammary vein or the external jugular vein. The blood was collected in 450 mL Terumo bags containing CPDA-1 (per 100 mL: Citric acid monohydrate, 0327 g; sodium citrate dihydrate, 2.63 g; monosodium phosphate dihydrate, 0251; dextrose anhydrous, 2.90 g; adenine, 0.0275, water for injection q.s.). The bags were stored at +4 °C and used within 24 h of collection.

##### Double Centrifugation Method

All separation steps were performed under a horizontal laminar flow hood in aseptic conditions. To prepare the PRP, the blood was drawn into sterile tubes, each of 50 mL. The tubes were centrifuged in an ALC PK 130 R PROKEME centrifuge at 100× *g* for 30 min. This caused separation of the blood into its three basic components: red blood cells, ‘buffy coat’, and platelet-rich plasma. Because of differing densities, the layer of red blood cells was formed at the lowest level, the buffy coat comprised the middle layer and PRP the upper layer. The PRP and, partially, the buffy coat were collected and placed in new 50 mL tubes and centrifuged again at 1500× *g* for 10 min. After this centrifugation, a platelet pellet and a supernatant called platelet poor plasma (PPP) were obtained. Then, for each animal, the platelet pellet was resuspended with different volumes of PPP to produce a PRP with a platelet level 3 times higher than the initial blood. The platelet count in the animals in the study varied from a minimum of 154 × 10^3^/µL to a maximum of 602 × 10^3^/µL. Platelet and leukocyte counts were performed on peripheral blood, PRP and final PRP, using an automatic impedance hematology analyzer HeCo Vet SEAC.

This PRP, prepared for mastitis treatment in a heterologous way, was subjected to bacteriological examination to verify its sterility and, after that, it was stored in 5 mL syringes. To allow the release of platelet factors, it was frozen at −80 °C and thawed at 37 °C three times to permit the breakdown of platelets as reported by Zimmermann et al. [23], then it was stored at −20 °C until its use.

#### 2.2.2. Statistical Analyses

The collection efficiency of platelets or yield for each PRP obtained by the double centrifugation tube method was analysed using the following formula [24]:Efficiency for platelet collection or yield = platelet count/µL in PRP × volume of PRP
platelet count in whole blood/µL X volume of whole blood

The Pearson correlation coefficient among PLT, YIELD and the initial WBC concentration was estimated using PROC CORR of SAS.

The yield was analyzed using the PROC GLM of SAS 9.4 (SAS Inc., Cary, NC, USA). The model included the fixed effects of the physiological state of the bovine (STATE: pregnant or not pregnant), SEASON (autumn or summer), CALVING INTERVAL (interval from last delivery to moment of blood collection), site of collection (SITE: jugular or mammary vein), AGE and number of BIRTHS. Platelet count (PLT) was also considered in the model as a covariate. The number of births were pooled into three classes (1–2, 3–4 and 5–6–7); also, three classes were Used to group the calving interval (months) between the last birth and the sampling of blood (≤3, 4–5, ≥6). Least square (LS) means were separated by pair-wise t-test and Tukey’s adjustment was applied. The mean separation for the main effects were performed on least square means using the PDIFF option of SAS. The distribution of the dependent variable is not normal, even after suitable transformations. However, as reported by Petrie and Watson [25], if the distribution is extremely different from normal, a size between 50 and 100 observations is required. In our study, having 135 cows, the distribution of the variable under analysis does not deviate excessively from normality (W = 0.98, *p* = 0.04; skewness = −0.043; kurtosis = −0.363) and our data could be analyzed using robust parametric tests such as ANOVA. Statistical differences were declared at *p* < 0.05.

## 3. Results

In bovine species, the physiological range of the platelet count is 412–1003 (×10^3^/µL).

The percentage yields of PRP obtained from the blood of enrolled cows varied from 1.04% (the lowest) to 24.5% (the highest). The various yields obtained were arbitrarily divided as indicated in Table 1. In blue, the high yield (>15%); in red, the medium-high yield (10–15%); in green, the medium-low yield (5–9%) and in violet the low yield (<5%).

The highest PRP yield was 24.5% in a 5-year-old animal that was not pregnant but had delivered three calves. From this animal, the blood was collected 2 months after its last parturition. The cow with the lowest yield (1.04%) was pregnant and heading towards the dry phase, being 8 months since its last parturition. It was also 5 years old and had delivered three calves.

The linear model shows that all variables had a statistical effect on the yield of PRP, except the season of collection.

### 3.1. Platelet and White Blood Cell Concentrations

Platelet concentrations ranged from a minimum of 154 × 10^3^/µL to a maximum of 602 × 10^3^/µL (the average ± standard deviation was 348.46 × 10^3^/µL ± 93.47 × 10^3^/µL). In all animals, the final PRP had a platelet level three times higher than the initial blood sample with a minimum of 462 × 10^3^/µL to a maximum of 1806 × 10^3^/µL (the average ± standard deviation was 1045.07 × 10^3^/µL ± 280.42 × 10^3^/µL).

The PRP yield was positively correlated with the platelet count in peripheral blood (r = 0.76, *p* < 0.0001).

White blood cell (WBC) concentrations in the initial blood samples ranged from a minimum of 1.17 × 10^3^/µL to a maximum of 13.09 × 10^3^/µL. In the final PRP, the white blood cells had a range from a minimum of 3.44 × 10^3^/µL to a maximum of 10.01 × 10^3^/µL.

A positive correlation was found between the initial WBC concentrations and WBC concentrations in the PRP (r = 0.32, *p* < 0.0001).

The PRP yield was not correlated with the initial WBC concentration (r = −0,11, *p* = 0.19).

### 3.2. Physiological Stage

The physiological stage in this study refers to the presence or absence of pregnancy. All pregnant animals gave a yield production of <10% while none of the nonpregnant cows gave a yield of <10% (Figure 1A).

A statistically significant difference (*p* = 0.0396) was found between the two compared physiological conditions with respect to the percentage yield of PRP which was higher in nonpregnant animals (13.67% ± 1.45) compared to pregnant animals (9.87% ± 1.28) (Figure 1B).

### 3.3. Season of Collection

The 135 cows were examined in two different seasons, summer and autumn. Figure 2A shows that low and medium-low yields were found in both seasons. High yields were distributed mainly in summer seasons.

The analysis of the data does not show statistically significant differences (*p* = 0.1865) between blood collection in summer (12.31% ± 5.21) or in autumn (13.22% ± 4.37) (Figure 2B).

### 3.4. Interval between the Last Birth and Sample Collection

The interval (expressed in months) between the last calving (or the beginning of lactation) and the collection of the blood sample was considered in this study.

Figure 3A shows that cows with a calving/picking interval of <3 months provided the highest yield percentages. Intermediate yield values are distributed over all the interval ranges, while percentages lower than 5% were only seen in cows with a calving/sampling interval between 4 and 5 months. The influence of the delivery/sampling interval on the yield is supported by the statistical analysis, through which a statistically significant difference (*p* = 0.0278) in the yield levels obtained between the interval of <3 months and the other two intervals was highlighted (14.61% ± 3.15 vs. 8.15% ± 2.39 and 9.39% ± 4.16) (Figure 3B).

### 3.5. Site of Blood Collection

Blood samples were collected from two different sites. In the case of jugular vein sampling, there is an equal distribution of animals in different yields of PRP (Figure 4A).

The analysis of these data reveals a statistically significant difference (*p* = 0.0031) between the two sampling sites, as evident in Figure 4B (11.37% ± 2.16 vs. 14.16% ± 2.28 in jugular vein and mammary vein, respectively).

### 3.6. Age of Cow

Only in the intervals 49–72 and 73–89 months was the yield of PRP < 5%. The higher, medium and medium-low rate were registered in the interval of 24–48 and 49–72 months (Figure 5A). The statistical analysis revealed a statistically significant difference between the age groups analyzed (*p* = 0.005; Figure 5B and Table 2).

### 3.7. Number of Births

The number of pregnancies, grouped into three categories, show a higher yield production of PRP in categories of 1–2 and 3–4 births with only 6.7% of cows (9 cows) having had 5–7 partums (Figure 6A). The statistical analysis revealed a statistically significant difference between the different categories analyzed (*p* = 0.05; Figure 6B and Table 3).

## 4. Discussion

PRP is very simple to prepare, using the double centrifugation technique, and easy to store at −20 °C. In the laboratory, it undergoes three cycles of freezing and thawing to break the alpha-granules and make the growth factors available for heterologous treatment. After that, the PRP is delivered to the farmers or to veterinarians in thermal containers to be used for different kinds of treatments in animals. In the farm, PRP must be kept at −20 °C and thawed only at the time of its use. Therefore, the PRP is an effective therapeutic product, easy to produce, store and manipulate, making it a valid product for the treatment of mastitis in cows or for other diseases or lesions.

Since PRP is a blood product, it is important to identify suitable donor cows for the preparation of heterologous PRP so that this could be readily available on the farm when diseases occur.

Having optimal blood donors within a farm gives a safe source of blood, and consequently, high PRP yields in the event of pathologies. The PRP yield represents the millilitres of PRP obtained with respect to the initial blood volume. In our studies, the PRP always had a platelet level three times higher than the initial blood, but the final volume of PRP with respect to the initial blood volume was very variable. Based on our experience, a percentage yield equal to or greater than 10%, was considered high.

To have a high PRP yield, the donor blood should have a high platelet count, since there is a correlation between the initial platelet level and the PRP yield, while the initial concentration of WBC does not influence this yield.

This result suggests that the procedure used for concentrating platelets is highly efficient for blood samples with high platelet concentrations, but less if platelets counts are lower. The cause of this discrepancy is unknown, but we can presume that the density of platelets in peripheral blood may be the basis of different recovery yields using centrifugation methods. The density of platelets may be quite variable depending on the content of granules, degranulation and platelet size. Unfortunately, these data were not collected. Similarly, the present study focused on evaluating the different yields of PRP and not on the biological activity of the PRP recovered from different cows and we cannot exclude that PRP isolated from different cows may exhibit different therapeutic properties irrespective of yield. Of course, for future studies, a preliminary evaluation of blood values would help in choosing the better blood donor cows, but in our retrospective data this was not possible.

From our previous data relating to the treatment of mastitic cows on different farms with heterologous PRP, we wanted to determine if factors other than the platelet count affected the final platelet content and PRP yield. The collected data related to the hygiene–health management of the various farms, the management of internal and external biosecurity, and the state of health and degree of welfare of each animal. Following the data collection, it became clear that some parameters were not suitable for inclusion in the analysis of factors influencing the yield of the PRP. This is because, within the sample of the dairy cows examined, the data relating to the hygiene of the bedding, the material of the bedding, the presence of cooling systems, the type of feeding, the type of housing, the degree of cleanliness of the animal and nutritional status, had characteristics common to all the farms in the study. This was perhaps to be expected since the farms all belong to the same production area.

However, there was more variability in other parameters, related to the individual cow characteristics. Specifically, the physiological stage, the season of blood collection, the interval between the last birth and the time of blood collection, the site of blood collection, the age of the cow and the parity, which were analysed for their influence on the PRP yield.

The data were difficult to examine since this study is a post-analysis of data collected during the mastitis treatment of different cows on different farms. The randomization in the selection of the cows as donors of PRP for mastitis treatment created a lack of homogeneity for each of the parameters analysed, such that there was not an equal number of subjects in each subcategory considered in this study. This generated a disproportion in the statistical analysis. This is particularly true for the number of nonpregnant cows which were much more represented than pregnant ones. The cows to be used as donors were selected by the farmers who preferred to avoid sampling pregnant cows in order to prevent stress. We cannot exclude that this fact may contribute to create a statistical bias.

The interval from delivery to blood collection had a statistically significant influence on the yield of PRP. The interval considered in this study varied from two weeks (the shortest period) to a maximum of eight months (the longest period) from delivery. The analysis showed that the cows with the highest PRP yield (equal to or above 10–15%) had an interval between the last calving and the time of collection of two weeks and 3 months, i.e., those cows in the early phase of lactation are better donor candidates. Percentages of <5% were only seen in cows with a calving–blood collection interval equal to or greater than 4 months. A lactation stage of up to 3 months appears to be necessary to obtain optimal yield levels. Jonsson et al. [26] comparing haematological parameters between pre-calving (samples were collected between 32 and 5 days) and post-calving (sampling occurred between 11 and 18 days after parturition) showed that in post-calving cows there is an increased platelet count compared to pre-calving cows (396 vs. 126 × 10^3^/µL). A higher platelet concentration in post-calving cows would explain a higher yield of PRP in the first three months of lactation (considering the positive correlation between the platelet level and PRP yield found in our statistical analysis). In addition to the increased platelet count in the blood, it is possible that the period of great stress induced by delivery, the negative energy balance, the lipomobilization and the availability of non-esterified fatty acids (NEFA), change the density of platelets, allowing a higher platelet recovery during the technical phase of centrifugation. It is possible that, due to the higher density of platelets, fewer end up in the buffy coat layer or in the lowest erythrocyte layer. This phenomenon has already been demonstrated in bovine WBC [27] but, to the best of our knowledge, no studies supporting changes in densities of platelets are available. Also, we cannot exclude that these platelets may be less brittle and may break less during centrifugation and manipulation. Indeed, among the cows examined, the highest and the lowest yields of PRP were 24.5% and 1.04% respectively, and both these animals were not pregnant, were 5 years old and had given birth three times, but the one that gave the highest yield had a 2-month interval from delivery to blood collection, while the one with the lowest yield had an 8-month interval from delivery to blood collection.

Similarly, the influence of the physiologic stage, meaning the presence or absence of pregnancy, was statistically significant. Insufficient PRP yield was only seen in pregnant cows, and nonpregnant cows appear to be better donors. These data confirm our previous results about post-calving cows and concur with Jonsson’s study [26] that nonpregnant cows have higher platelet counts and provide a better PRP yield.

Regarding the age of cows, and indirectly their parity, the statistical analysis shows a statistically significant difference in the subcategories, and the distribution of these data seem to indicate that cows between 49 and 72 months old with 3–4 births are good donor candidates. In human medicine, Berger et al. [28] observed an age-dependent optimal platelet lysate concentration from aged but not from young donors. The platelet lysates from aged donors were more effective in promoting tenocyte proliferation and migration in vitro because they had higher platelet counts and platelet-derived protein levels compared to young donors [28]. There is no information in the literature relating to platelet counts with advancing animal age and, from our statistical analysis, there is no correlation between the platelet level and age. In our study, the few animals 95–120 months old gave good yields of PRP (high or medium-high), but the low number of animals in this group meant it was not possible to identify a positive response in this age group.

Even the qualitative parameters, such as the site of collection had statistically relevant effects on the yields of PRP in this study. Most samples were taken from the mammary vein (89%) because the veterinary surgeon observed that cows were less stressed during the collection. Indeed, cows do not like to be manipulated on the front and are used to be handled in the back, also related to milking procedures. We do not know if the higher yield from the mammary vein could be linked to the fact that the density of the platelets is different in the mammary blood and, therefore, gives a greater collection during the centrifugation procedures. Obviously, the collection of blood from the mammary vein could be related to mammary vein thrombosis or phlebitis, persistent unilateral mammary oedema, and endocarditis, but within our experience we have never had any unfavourable events. Clearly, the veterinarian must have considerable skill and experience in carrying out this type of sampling, which must be performed in perfect hygienic conditions to avoid infections.

There was no statistically significant difference in the yields obtained in the two seasons.

## 5. Conclusions

These data seem to suggest a hypothetical profile that should provide a good donor cow, and their blood should be able to provide PRP yields above the 10% threshold considered optimal. The donor cow, like the human donor, must be in good health, free from infectious diseases and management stresses. This study highlights some features that allow us to define a cow as a good donor: it must not be pregnant but should be within the first three months after giving birth, be between 49 and 72 months of age, with 3–4 parity, and it is preferable that its blood is collected from the mammary vein. From collected data, we have estimated that in every farm where we have carried out mastitis treatments with PRP, there are always about 5–10% of animals with these characteristics and these animals are sufficient to represent a good source of blood donors to produce PRP for heterologous use in each farm. Further studies on the functionality of recovered platelets are needed to confirm which are the characteristics that can select the best donors for PRP to create a pool of high-quality platelets stored on farms for rapid use.

## Figures and Tables

**Figure 1 animals-11-02296-f001:**
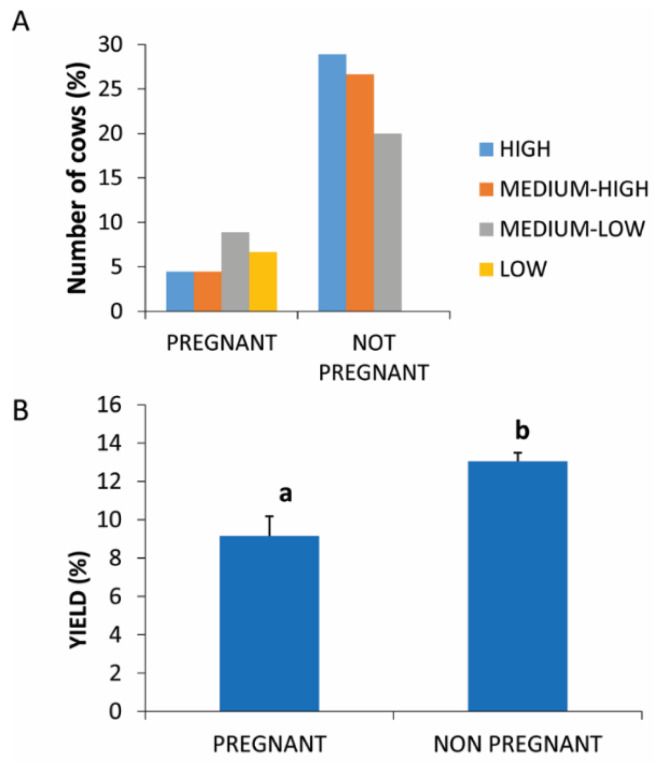
PRP yield in pregnant and not pregnant. (**A**) Distribution (%) of pregnant and not-pregnant cows in the different PRP yield classes. (**B**) Comparison of PRP yield in pregnant and not-pregnant cows. Values are expressed as means and standard deviations. Different letters represent statistically significant differences.

**Figure 2 animals-11-02296-f002:**
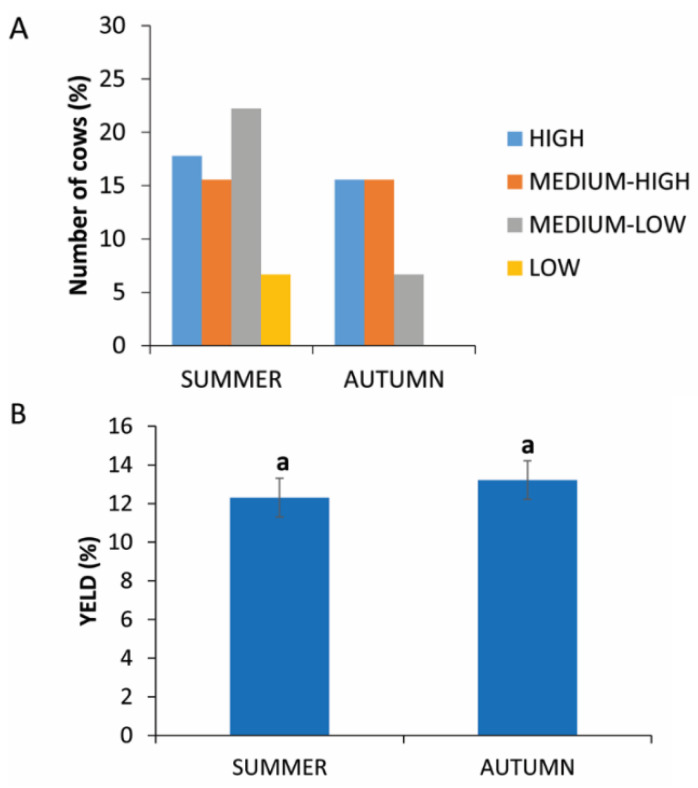
PRP yield in two different seasons. (**A**) Distribution (%) of cows in the different PRP yield classes in autumn or summer. (**B**) Comparison of PRP yield in autumn or in summer. Values are expressed as means and standard deviations. Different letters represent statistically significant differences.

**Figure 3 animals-11-02296-f003:**
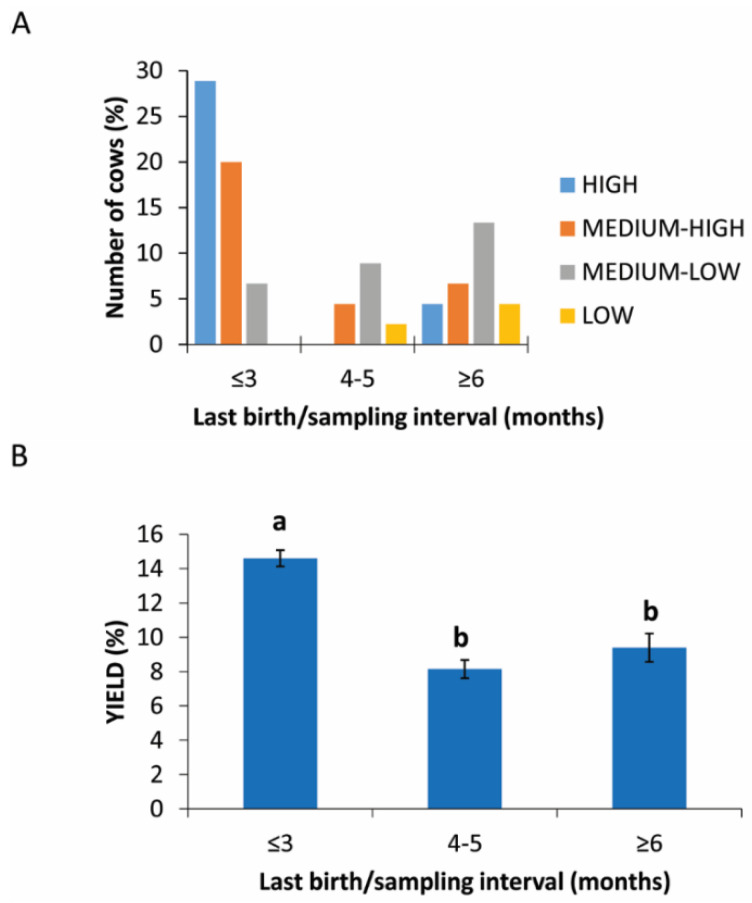
PRP yield in different intervals between the last birth and sample collection. (**A**) Distribution (%) of cows in different last birth/sampling intervals expressed in months in the different PRP yield classes. (**B**) Comparison of PRP yield among different groups based on last birth/sampling intervals expressed in months. Values are expressed as means and standard deviations. Different letters represent statistically significant differences.

**Figure 4 animals-11-02296-f004:**
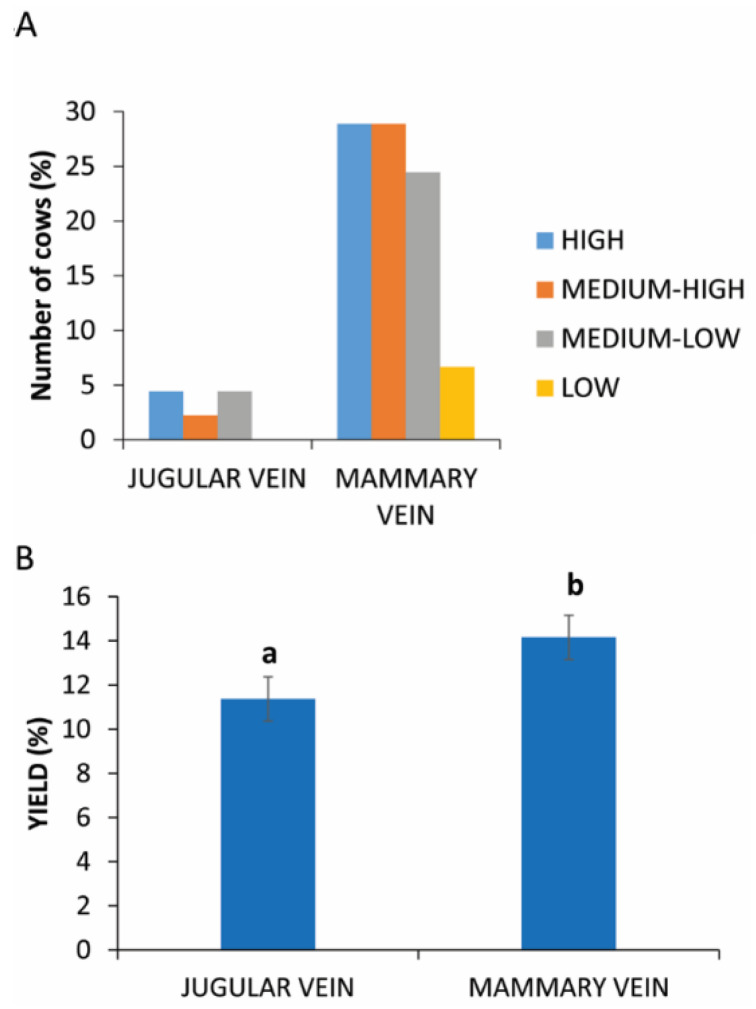
PRP yield between two different sites of blood collection. (**A**) Distribution (%) of cows in different PRP yield classes considering the collection in mammary or jugular vein. (**B**) Comparison of PRP yield between two different sites of blood collection. Values are expressed as means and standard deviations. Different letters represent statistically significant differences.

**Figure 5 animals-11-02296-f005:**
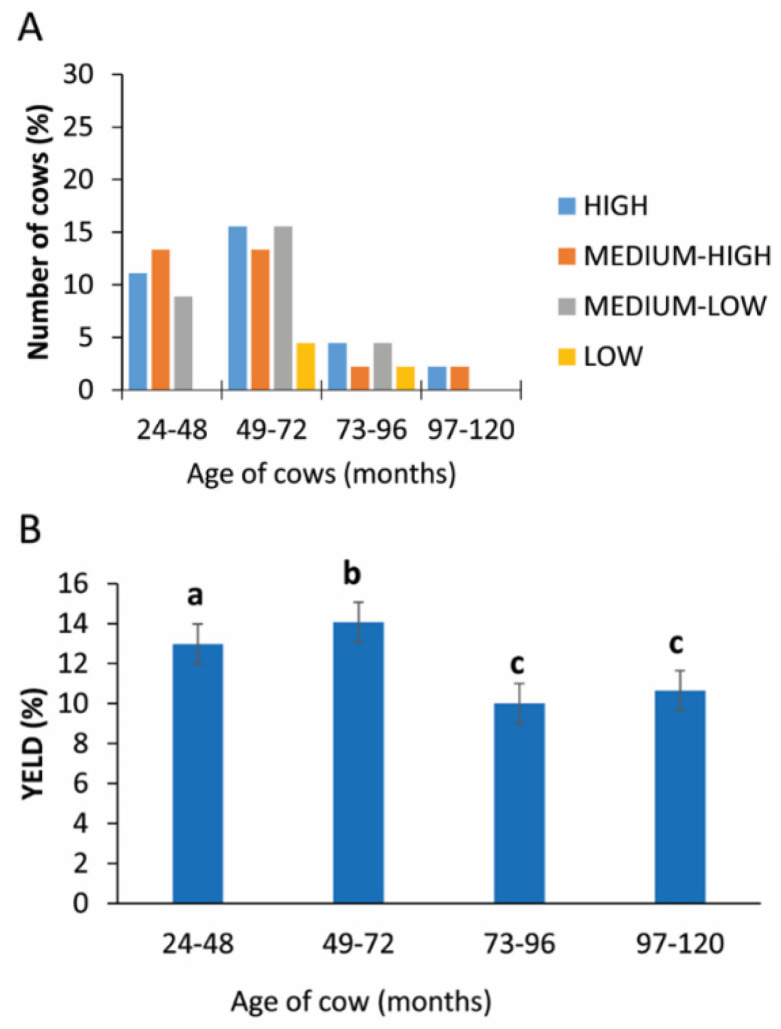
PRP yield considering the age of cow. (**A**) Distribution (%) of cows in different PRP yield classes considering the different age classes. (**B**) Comparison of PRP yield in different age classes. Values are expressed as means and standard deviations. Different letters represent statistically significant differences.

**Figure 6 animals-11-02296-f006:**
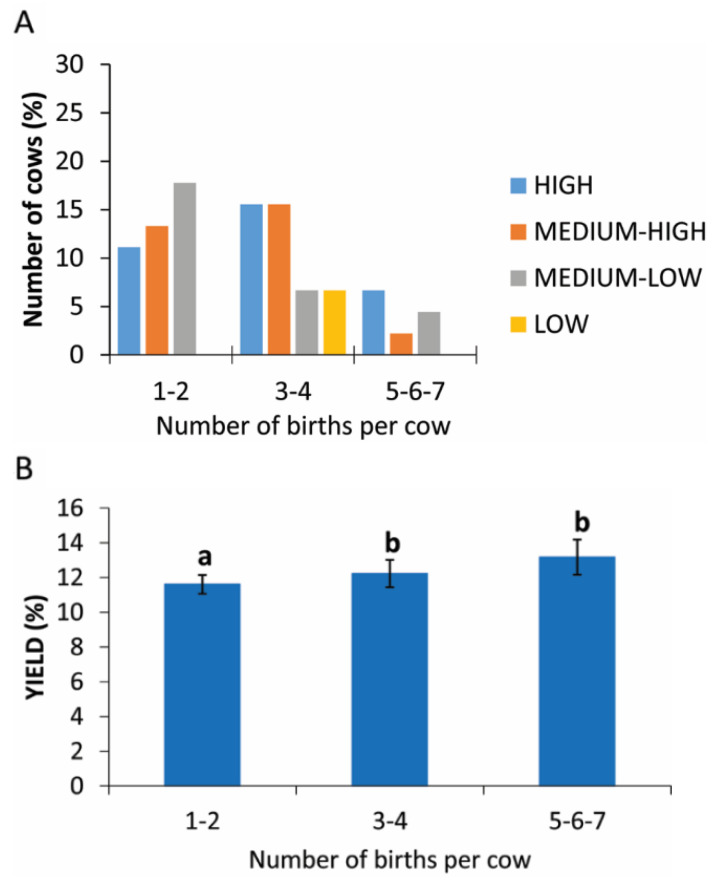
PRP yield considering the number of births. (**A**) Distribution (%) of cows in different PRP yield classes considering the number of births. (**B**) Comparison of PRP yield considering the number of births. Values are expressed as means and standard deviations. Different letters represent statistically significant differences.

**Table 1 animals-11-02296-t001:** Color legend of different levels of percentage yield of PRP.

Level	% Yield
High	>15
Medium-high	Between 10–15
Medium-low	Between 5–9
Low	<5

**Table 2 animals-11-02296-t002:** Yield of PRP in different ages of cows.

Age of Cows (Months)	Yield (%)
24–48	12.98 ± 3.19
49–72	14.07 ± 2.58
73–96	10.01 ± 1.72
97–120	10.56 ± 1.25

Legend: data are expressed as mean and standard deviation.

**Table 3 animals-11-02296-t003:** Yield of PRP in cows with different numbers of births.

Number of Births per Cow	Yield (%)
1–2	11.56 ± 2.06
3–4	13.93 ± 2.13
5–6–7	10.93 ± 2.30

Legend: data are expressed as mean and standard deviation.

## Data Availability

All datasets generated for this study are included in the article.

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
