# Peer review of "Physiological Parameters to Identify Suitable Blood Donor Cows for Preparation of Platelet Rich Plasma"

_animals, 2021, doi:10.3390/ani11082296_

Round 1
Reviewer 1 Report
Overall, this is an interesting study and would complement the previous pertinent literature. However, the paper needs further work before it can be accepted for publication. In my opinion, methods need a major revision.
My major concern is about the structure of methods. The study is structured as prospective research, but it is effectively a retrospective analysis of data. In my opinion, methods should describe what the Authors did with the records collected during the previous research, during which they went really to the farms. For this study, data previously collected were analyzed for different aims and statistically evaluated. Statistical analysis, for this reason, is the hearth of the study and should be analytically described step by step.
Overall, this is an interesting study and would complement the previous pertinent literature. However, the paper needs further work before it can be accepted for publication. In my opinion, methods need a major revision.
Introduction:
- Lines 61-69: lines 61 to 68 show the indication and clinical use of PRP in human medicine which largely fall within the disciplines shown in lines 68-69. In my opinion, the paragraph could be better structured and summarized.
- Lines 89-90: this is true exclusively for Italian law
Materials and methods:
I understood that this is a retrospective evaluation of data collected during previous research. Therefore, materials and methods should describe from which database, forms, or records data were obtained and in detail the methods of analysis. Collecting data on farms is part of the previous study. Characteristics of the farms and animals should be reported in the results, as resulting from the examen of the previous records, unless you have returned to the farms to obtain new data.
Lines 198-206: in my opinion, this is the heart of the study, and it needs a more extensive description. Which PROC GLM was used? 9.2? Did you use an unbalanced ANOVA or a Quadratic Least Squares Regression to analyze the GLM results?
An “a posteriori” Power analysis should be included. In the discussion, the distribution of data was discussed, but in my opinion, you should describe here if you analyze or not the distribution of data and why. I am sorry, but I do not routinary use PROC GLM, and perhaps for this reason I do not understand very well, but the description of statistical analysis should include the indication of the single steps of the analysis because not all the readers have to be aware of a specific software.
Furthermore, in the results the correlation between platelet yields and hematological parameters is reported, but in statistical analysis it was not mentioned (lines 226-233).
Results:
- Lines 226-233: see above, what was said about statistical analysis.
Discussion:
- Lines 292-299: the evaluation of the advantages of the use in field of PRP is not an objective of this study and the sentence should be deleted.
- Lines 345-349: as said above, something about the evaluation of the distribution of data should have been indicated in the paragraph of the statistical analysis.
Tables 2 and 3 legends: please, you should indicate what are the percentages reported, means and St. dev?
Author Response
Introduction:
- Lines 61-69: lines 61 to 68 show the indication and clinical use of PRP in human medicine which largely fall within the disciplines shown in lines 68-69. In my opinion, the paragraph could be better structured and summarized.
Answer: the authors thank the referee for her/his comment, but we decide not to summarize this part because the other referees asked us to deepen it
- Lines 89-90: this is true exclusively for Italian law
Answer: the authors thank the referee for her/his comment. The specification about the Italian law has been added at line 77.
Materials and methods:
I understood that this is a retrospective evaluation of data collected during previous research. Therefore, materials and methods should describe from which database, forms, or records data were obtained and in detail the methods of analysis. Collecting data on farms is part of the previous study. Characteristics of the farms and animals should be reported in the results, as resulting from the examen of the previous records, unless you have returned to the farms to obtain new data.
Lines 198-206: in my opinion, this is the heart of the study, and it needs a more extensive description. Which PROC GLM was used? 9.2? Did you use an unbalanced ANOVA or a Quadratic Least Squares Regression to analyse the GLM results?
Answer: the authors thank the referee for her/his comment. PROC GLM SAS 9.4 (SAS Inc., Cary, NC) has been used. The GLM procedure has been used for performing the analysis of variance (ANOVA) for unbalanced data as in this study.
An “a posteriori” Power analysis should be included.
Answer: the authors thank the referee for her/his suggestion. The confidence interval of all effects’ size, with the exception of the season, did not contain “0,” indicating “statistical significance,” then a posteriori analysis is not necessary.
In the discussion, the distribution of data was discussed, but in my opinion, you should describe here if you analyse or not the distribution of data and why.
Answer: the authors thank the referee for her/his suggestion. We agree with the reviewer, the sentence has been moved to M&M at line 184-189.
I am sorry, but I do not routinary use PROC GLM, and perhaps for this reason I do not understand very well, but the description of statistical analysis should include the indication of the single steps of the analysis because not all the readers have to be aware of a specific software.
Answer: the authors thank the referee for her/his comment. As it is a widely used and well-known statistical method, the description of the individual steps is hardly ever repeated. Generally, it is sufficient to specify the type of model and the factors considered, as can be seen in ‘M&M’ section of many research papers. In the manuscript under revision, we have indicated the categorical factors and the covariate.
Furthermore, in the results the correlation between platelet yields and hematological parameters is reported, but in statistical analysis it was not mentioned (lines 226-233).
Answer: the authors thank the referee for her/his suggestion. We agree and a sentence regarding the correlation coefficient has been included at lines 174-175.
Results:
- Lines 226-233: see above, what was said about statistical analysis.
Answer: the authors thank the referee for her/his suggestion. We have already answered.
Discussion:
- Lines 292-299: the evaluation of the advantages of the use in field of PRP is not an objective of this study and the sentence should be deleted.
Answer: the authors thank the referee for her/his comments. Any reference to the use in field has been deteled.
- Lines 345-349: as said above, something about the evaluation of the distribution of data should have been indicated in the paragraph of the statistical analysis.
Answer: the authors thank the referee for her/his suggestion. We have already answered.
Tables 2 and 3 legends: please, you should indicate what are the percentages reported, means and St. dev?
Answer: the authors thank the referee for her/his comments. The percentage represents mean and standard deviation as has been specified in each caption.
Reviewer 2 Report
Allover comments:
-The simple summary and the abstract are too close to each other. The simple summary should contain more information for non-veterinarians.
-The manuscript describes the best option (cows, season etc.) to gain platelet rich plasma. This manuscript does not describe mastitis therapy, but data of PRP are extracted from the publication with the subject of mastitis therapy. The difference between these two issues
- best cows to produce PRP
- mastitis therapy
is not clearly made. The reviewer would recommend to focus on the production of the best possible PRP. Once it is clear which cows to chose to produce the best possible PRP, a new prospective study on mastitis including more data could be performed.
-Mammary vein: The mammary vein should not be used for either blood sampling or drug administration because complications of mammary venipuncture may have disastrous results, such as mammary vein thrombosis or phlebitis, persistent unilateral mammary edema, and endocarditis. In general, it is contraindicated to use the mammary vein therapeutically unless the cow has a life-threatening illness and is in a compromised position, such that the jugular vein is inaccessible.
Specific comments:
How many ml of PRP can be gained from 450ml whole blood? Does 10% mean 45ml?
Line 63: The anti-inflammatory aspect was unfortunately only seen in a case-series of 8 women with face-lifts. Other literature proving this fact?
Lines 73-76: Is this 1 study enough to prove efficacy of PRP in mastitis?
Line 77: indicate reference to "extensive studies"
Lines 93-104: again the present study seems to be interwoven and dependent on the study of Lange-Consiglio in 2014. The reviewer does not understand, why treatment of mastitis was included and not only blood parameters of cows. The question is: how to prepare the best PRP? Age of the cows, time post partum, season, vena etc.
Lines 97-99: is the number of the animal experimentation allowance?
Line 112: what means "without overcrowding"? (nr of animals per square meter)
Line 115: what means "hygiene good/excellent" (criteria?)
Line 116: "BCS good" means? (values? ranges? loss?)
After chosing cows: did you analyze minimal blood values as hematocrit, WBC etc. before taking 450ml of blood? (1.17 x 10*3/microl is very low!). How was health state of cows defined?
Table 1: "10" appears in 2 categories
Lines 218-220: please indicate months in brackets. Was hemoconcentration present in summer?
Lines 248-260: The reviewer would not recommend the mammary vein neither for medical treatment nor for blood taking, as all complications with mammary veins are fatal.
Lines 261-264: added=99.6%
Author Response
The Authors thanks the Reviewers for considerations and helpful suggestions. According to the comments and suggestions, we have carefully evaluated all critical points and the manuscript has been thoroughly revised. The Authors hope that now the manuscript is suitable for publication on “Animals”.
REFEREE 1
All-over comments:
-The simple summary and the abstract are too close to each other. The simple summary should contain more information for non-veterinarians.
ANSWER: the authors thank the referee for her/his comment and the simple summary has been modified
-The manuscript describes the best option (cows, season etc.) to gain platelet rich plasma. This manuscript does not describe mastitis therapy, but data of PRP are extracted from the publication with the subject of mastitis therapy. The difference between these two issues
- best cows to produce PRP
- mastitis therapy
is not clearly made. The reviewer would recommend to focus on the production of the best possible PRP. Once it is clear which cows to choose to produce the best possible PRP, a new prospective study on mastitis including more data could be performed.
ANSWER: the authors thank the referee for her/his comment and modified the introduction focusing only the identification of the good blood donor cow
-Mammary vein: The mammary vein should not be used for either blood sampling or drug administration because complications of mammary venepuncture may have disastrous results, such as mammary vein thrombosis or phlebitis, persistent unilateral mammary edema, and endocarditis. In general, it is contraindicated to use the mammary vein therapeutically unless the cow has a life-threatening illness and is in a compromised position, such that the jugular vein is inaccessible.
ANSWER: the authors thank the referee for her/his comment. The authors fully agree with the referee but from the blood samples taken from the jugular vein the yields were very low. Despite not having evaluated the ACTH values, apparently the animal was more stressed, and the platelet value obtained was lower than when from the same animal the blood was collected from the mammary vein.
During our experience we have never had any unfavourable events such as phlebitis which can occur if the mammary vein is used for the administration of drugs. The release of drugs from the mammary vein could then cause phlebitis but in our case, we perform a blood sample and not drug administration.
The sampling from the mammary vein allows a containment without stress, obviously, the veterinarian must have considerable skill and experience in carrying out this type of sampling which must be performed in perfect hygienic conditions to avoid infections.
Specific comments:
How many ml of PRP can be gained from 450ml whole blood? Does 10% mean 45ml?
ANSWER: the authors thank the referee for her/his question. The yield of PRP are the milliliters of PRP obtained respect to milliliters of initial blood as specify at line 77-78 of the manuscript, then 10% means 45 ml from 450 ml of initial blood.
Line 63: The anti-inflammatory aspect was unfortunately only seen in a case-series of 8 women with face-lifts. Other literature proving this fact?
ANSWER: the authors thank the referee for her/his comment. Some papers evidenced the anti-inflammatory effect of PRP. Liu et al. (2007), in an in vitro study evaluated that PRP promotes significant changes in monocyte-mediated proinflammatory cytokine/chemokine release, suggesting that PRP may suppress cytokine release, limit inflammation, and, thereby, promote tissue regeneration.
Sundman et al. (2014) evidence that anti-inflammatory activities of PRP support its use in osteoarthritis joints to reduce pain and modulate the disease process.
Ameer et al. (2018) observed a great role of PRP in the periodontal treatment by its anti-inflammatory effect.
All these papers have been added in the manuscript.
Lines 73-76: Is this 1 study enough to prove efficacy of PRP in mastitis?
ANSWER: the authors thank the referee for her/his comment. In the paper of Lange-Consiglio et al. (2014), 229 animals were enrolled. In addition, in the experimental European Union (PSR 2014-2020) project, 480 animals were enrolled on 20 farms. The results of this last project, that were not published, confirmed the results obtained in 2014 and the efficacy of this treatment for the mastitis in the acute and chronic pathology.
Line 77: indicate reference to "extensive studies”.
ANSWER: the authors thank the referee for her/his comment. As previously explained, there is only one paper published in 2014. Extensive studies were related to the numerous animals treated in the European project whose data are not published. After these considerations, in any case, the word "extensive" was removed.
Lines 93-104: again, the present study seems to be interwoven and dependent on the study of Lange-Consiglio in 2014. The reviewer does not understand, why treatment of mastitis was included and not only blood parameters of cows. The question is: how to prepare the best PRP? Age of the cows, time post-partum, season, vena etc.
ANSWER: the authors thank the referee for her/his comment. The manuscript has been modified and the correlation with the mastitis treatment has been mitigated.
Lines 97-99: is the number of the animal experimentation allowance?
ANSWER: the authors thank the referee for her/his comment. The number cited in the paper is the project’s number that allowed us to extend the study of the PRP treatment in the mastitis and to confirm the data of 2014. Thank you to these data, we have collected information to define the good blood donor cow.
Line 112: what means "without overcrowding"? (nr of animals per square meter)
ANSWER: the authors thank the referee for her/his comment. Each animal had a well-defined location available, so no overcrowding was present.
Line 115: what means "hygiene good/excellent" (criteria?)
ANSWER: the authors thank the referee for her/his comment. The hygiene criteria were mainly related to the milking methods. It was important that the selected farms performed properly pre- and post-dipping. This has been specified in the manuscript.
Line 116: "BCS good" means? (values? ranges? loss?)
ANSWER: the authors thank the referee for her/his comment. The BCS must be calculated at least in 4 moments: 1. Delivery; 2. First insemination; 3. Half lactation; 4. Drying.
The range of a good BCS is between 2.75 and 3.50 (AHDB Dairy, 2012; Joner et al., 2016). To enrolle animal in the study, only animals with good BCS depending on physiological stage were enrolled.
After chosing cows: did you analyze minimal blood values as hematocrit, WBC etc. before taking 450ml of blood? (1.17 x 10*3/microl is very low!). How was health state of cows defined?
ANSWER: the authors thank the referee for her/his comment. Minimal blood values were evaluated only after the blood collection from the animals that were chosen based on their anamnestic and clinical evaluation. The criteria adopted for the choice were: absence of pathologies, non-use of drugs for at least 30 days, good BCS, average milk production typical of the animal depending on the physiological stage and general clinical evaluation. As stated in the manuscript the aim of the work is to evaluate retrospectively the PRP yield obtained from animals whose blood status were not previously known but that were chosen only based on anamnestic and clinical parameters. So, the blood values like haematocrit or others were not used to define the health status of the cow. WBC value of 1.17 x 10*3/microl are indeed indicative of severe neutropenia. Neutrophil numbers decrease in the postpartum period, and lactating cows have lower WBC counts than nonlactating cows. With automated cell counters, falsely low WBC counts might occur due to clumping of leukocytes, and falsely high WBC counts might be caused by nucleated red blood cells, insufficient lysis of erythrocytes, excessive Heinz bodies, or clumping of platelets.
Our lowest platelet count in the blood sample was 154x10*3/microl that is a below average platelet count but often found in the animals we analyzed.
Of course, a preliminary blood values evaluation would help in choosing the better blood donor cows but in our retrospective data this was not possible.
Table 1: "10" appears in 2 categories.
ANSWER: the authors thank the referee for her/his comment and apologize for the mistake. Values have been changed.
Lines 218-220: please indicate months in brackets. Was hemoconcentration present in summer?
ANSWER: the authors thank the referee for her/his suggestion. The months have been added. Hemoconcentration in the initial blood sample was evaluated and showed no particular fluctuation. Moreover, we were mainly interested in analysing the parameters that more influence the PRP efficacy that are platelet count and WBC count for the antibacterial activity that some PRP exerts.
Lines 248-260: The reviewer would not recommend the mammary vein neither for medical treatment nor for blood taking, as all complications with mammary veins are fatal.
ANSWER: the authors thank the referee for her/his suggestion, and we agree. We have already answered to this question.
Lines 261-264: added=99.6%
ANSWER: Authors apologize but this comment is not clear.
Reviewer 3 Report
The study is interesting and brings a practical approach in the treatment of mastitis in cows. However, the study lack in details and is confuse many times. The manuscriptshould be carefully reviewed and reorganized for clarification. The Methodology are lacking and must be improved. Many aspects are confused. Forexample, the PRP after the second centrifugation, how was it recovered? L132-143 –the authors mentioned that after the second centrifugation (1500 xg/10 min) the platelet pellet was suspended with the supernatant or platelet poor plasma (PPP).
What is the difference between the supernatant and the PPP? Also, each pellet wasresuspended in how many mL of PPP? The PRP preparation should be clarified.
How much volume of PRP was obtained after processing?
L144-146 – “The PRP was stored in 5 ml syringes and, to release platelet factors, it wasfrozen at -80°C and thawed at 37°C three times [19]. The PRP was subjected tobacteriological examination to verify its sterility.” Why was it performed? Was the PRP used for treatment? If yes, the PRP was submitted to three freezing-thawing cycles to
releaseand immediately used or frozen for later? Where are the methodology and
results?How was the PRP evaluated?
Results are confused for understanding and this section is mixed with parts of discussion. The authors missed parts of the methodology in the Results section. It is difficult to understand the results and the figures. Example:
L193 – “All pregnant animals gave a yield production < 10% and none of the non-pregnant cows gave a yield of < 10% (Figure 1A).” But in the figure 1A there arepregnant cows in all groups “high yield (> 15%); medium-high yield (10-15%); medium-low yield (5-10%) and low yield (<5%).”
The authors did not provide the final platelet concentration in the PRP (mean±SD and
range).
The authors should split and provide the number of animals in each group to beevaluated in the methodology, and only provide the results of the analyses in theResults section.
Also, if the percentage of animals in each group is not similar, the results and figures must be adjusted. Example Figure 1A (L195) – 75% of cows are open and only 25% pregnant but the figure shows the absolute values, which make a false image. It should be change through the manuscript.
The authors must improve the legends of the figures.
Author Response
The Authors thanks the Reviewers for considerations and helpful suggestions. According to the comments and suggestions, we have carefully evaluated all critical points and the manuscript has been thoroughly revised. The Authors hope that now the manuscript is suitable for publication on “Animals”.
The study is interesting and brings a practical approach in the treatment of mastitis in cows. However, the study lack in details and is confuse many times. The manuscript should be carefully reviewed and reorganized for clarification. The Methodology are lacking and must be improved. Many aspects are confused.
For example, the PRP after the second centrifugation, how was it recovered? L132-143 –the authors mentioned that after the second centrifugation (1500 xg/10 min) the platelet pellet was suspended with the supernatant or platelet poor plasma (PPP). What is the difference between the supernatant and the PPP? Also, each pellet was resuspended in how many mL of PPP? The PRP preparation should be clarified.
How much volume of PRP was obtained after processing?
ANSWER: the authors thank the referee for her/his comment and apologize for the confusion. After the second centrifugation, it was obtained a supernatant called platelet poor plasma (PPP), then the supernatant and the PPP are the same thing. The mL used to resuspend the platelet pellet depending on the platelet concentration. For each animal, we performed the analyses of this value to calculate the mL of PPP to use to obtain a PRP with a concentration higher 3-5 times the initial value of platelet of the animal.
For each animal we obtained different volumes of PRP and for this reason we calculated the yield that depends on different factors.
L144-146 – “The PRP was stored in 5 ml syringes and, to release platelet factors, it was frozen at -80°C and thawed at 37°C three times [19]. The PRP was subjected to bacteriological examination to verify its sterility.” Why was it performed? Was the PRP used for treatment? If yes, the PRP was submitted to three freezing-thawing cycles to release and immediately used or frozen for later? Where are the methodology and results? How was the PRP evaluated?
ANSWER: the authors thank the referee for her/his comment and apologize for the confusion.
The PRP was used for heterologous mastitis treatment and thank you this study we have had data available relating to individual donor cows to identify a good blood donor. The PRP was prepared under a laminar flow hood horizontal in aseptic conditions, but its sterility was evaluated in any case to be sure to use a sterile product for the mastitis treatment. Then, the PRP was immediately submitted to three freezing-thawing cycles to release the growth factors and frozen at -20°C to use later. By the study of Zimmermann et al. (2003) we have verified the breakage of platelets and the release of growth factors (data not shown).
Results are confused for understanding and this section is mixed with parts of discussion. The authors missed parts of the methodology in the Results section. It is difficult to understand the results and the figures. Example:
L193 – “All pregnant animals gave a yield production < 10% and none of the non-pregnant cows gave a yield of < 10% (Figure 1A).” But in the figure 1A there are pregnant cows in all groups “high yield (> 15%); medium-high yield (10-15%); medium-low yield (5-10%) and low yield (<5%).”
ANSWER: the authors thank the referee for her/his comment and apologize for the confusion.
In the figure 1A the distribution (frequency) of the animal has been reported for each physiological status. In each physiological stage, the animals are distributed in animal with high, medium-high, medium-low and low yield. In the figure 1B, the average of PRP yield underlines that the pregnant animals had a yield less than 10% and non-pregnant animals a yield major than 10%.
This is the meaning of the figures for each studied variable.
In the revision, the methodology and the discussion have been removed from the Results section that now describe only the results obtained.
The authors did not provide the final platelet concentration in the PRP (mean ± SD and
range).
ANSWER: the authors thank the referee for her/his comment and apologize for the confusion.
In our study we produced PRP with a concentration of platelet 3 times higher than the physiological level.
At lines 198-199 we reported the range of platelet “Platelet concentrations ranged from a minimum of 154x103/µl to a maximum of 602x103/µl. In all animals, the final PRP had a platelet level higher 3 times than the initial blood sample” then the final platelet concentration in the PRP was from a minimum of 462x103/µl to a maximum of 1806x103/µl.
The means and the SD have been added in the manuscript.
The authors should split and provide the number of animals in each group to be evaluated in the methodology, and only provide the results of the analyses in the Results section.
Also, if the percentage of animals in each group is not similar, the results and figures must be adjusted. Example Figure 1A (L195) – 75% of cows are open and only 25% pregnant but the figure shows the absolute values, which make a false image. It should be change through the manuscript.
The authors must improve the legends of the figures.
ANSWER: the authors thank the referee for her/his comment. In the revision, the methodology has been removed from the Results section that now describe only the results obtained.
In the figures, no absolute values are reported but only the percentage (the absolute values are reported in the M&M in the bracket). The figures 5 and 6 have been modified and uniformed to the other figures
The legends have been improved too.
Round 2
Reviewer 2 Report
Thanks for changing your manuscript – it is much clearer now.
Just a couple of remarks:
Line 127 : please indicate literature for BCS including which scale you use
Discussion:
- Thanks for your answer, please add a statement on mammary vein in the discussion (at least pointing to the risks!)
- Please add a sentence in the discussion, that it would be favourable to first analyse blood values (hematocrit, platelets) before using donor cows
Explanation to reviewers' comment in the first version: adding the different age categories resulted in 99.6% instead of 100% (but you deleted this part anyway)
Author Response
Thanks for changing your manuscript – it is much clearer now.
Answer: the authors thank the referee for her/his comment
Just a couple of remarks:
Line 127 : please indicate literature for BCS including which scale you use
Answer: at line 108 a reference has been added
Discussion:
- Thanks for your answer, please add a statement on mammary vein in the discussion (at least pointing to the risks!)
Answer: at line 364-368 two sentences have been added
- Please add a sentence in the discussion, that it would be favourable to first analyse blood values (hematocrit, platelets) before using donor cows
Answer: at line 283-285 a sentence has been added
Explanation to reviewers' comment in the first version: adding the different age categories resulted in 99.6% instead of 100% (but you deleted this part anyway)
Answer: Thank you anyway for your suggestion
Reviewer 3 Report
The authors had included most of reviwers purposed modifications and now is suitable for publication in the present form.
Author Response
The authors thank the referee for her/his comment